# UniSpike: Boosting the Performance of Spiking Neural Networks with Hybrid Training

## Abstract

Spiking neural networks (SNNs) are increasingly studied for their brain-inspired computing paradigm, offering high efficiency and sparse activation. However, achieving high accuracy with a small time-step remains challenging for SNNs. In this paper, we propose UniSpike, a hybrid training framework that combines the *high-accuracy feature of ANN-to-SNN conversion algorithms* and the *ultra-low time-step inference feature of direct training algorithms*. UniSpike converts a quantized ANN into its SNN counterpart, then fine-tunes the converted SNN. To replace the SNN-unfriendly operators in the ANN, UniSpike proposes Unified-Clip. Unified-Clip is equivalent to spike neurons and can replace SNN-unfriendly operators (*i.e.* softmax, layernorm, and GeLU) *without degrading ANN accuracy*. With Unified-Clip, UniSpike proposes UniFormer, a novel transformer that is *addition-only and supports step-by-step inference*. UniFormer allows all the matrix multiplications except the patch embedding to be realized by simple addition and eliminates synchronization operators across time-steps. With UniFormer, UniSpike achieves 80.9% accuracy on ImageNet, outperforming the SOTA direct training algorithm Spike-Driven TransformerV2 (80.0%) with addition-only and step-by-step features with 4 time-steps. Compared to the SOTA conversion-based algorithm SpikeZIP-TF, UniSpike reduces 5.7× energy with comparable accuracy.

## 1 Introduction

Spiking neural networks (SNNs) *leverage spikes to encode, transmit, and learn information from the world*. Compared to artificial neural networks (ANNs), SNNs exhibit a more efficient computational paradigm, offering notable advantages such as addition-only computation, step-by-step inference pattern and *etc*. (Roy et al., 2019). However, *achieving both high accuracy and low inference time-steps while maintaining these features remains challenging*.

Existing SNN learning algorithms can be generally categorized in threefold: *direct training (DT)*, *ANN-to-SNN conversion (A2S)*, and *hybrid training (HT) (Rathi et al., 2023)*. The DT algorithm (Wu et al., 2019) trains SNN using the back-propagation-through-time (BPTT) algorithm (Werbos, 1990) with a surrogate gradient (Neftci et al., 2019). Nevertheless, *it suffers from lower task accuracy compared to ANNs due to inaccurate gradient approximation* (Neftci et al., 2019). A2S methods (Rueckauer et al., 2017; Hu et al., 2023; You et al., 2024; Wang et al., 2023) transfer the parameters of the pre-trained ANN into its SNN counterpart that yields close-to-ANN accuracy. However, *the converted SNN consumes high inference time-steps (64 time-steps) and massive synaptic operations (aka. #SOP) to achieve accuracy on par with ANN* (You et al., 2024; Wang et al., 2023).

By combining DT and A2S algorithms, *HT algorithms inherits the pretrained parameters of target ANN as initialization, then fine-tune it with specific BPTT to achieve close-to-ANN accuracy at ultra-low time-steps*. However, existing HT methods (Baltes et al., 2023; Abuhajar et al., 2025) face several challenges. Firstly, Integrate&Fire (IF) neuron is not strictly equivalent to quantized activation function (*e.g.*, quantized-ReLU) (Bu et al., 2023). The activation-neuron mismatch causes sub-optimal weight initialization, affecting the accuracy of SNN fine-tuning. Secondly, the existing lossless conversion work (You et al., 2024) uses an ANN-specific designed network (ViT-base) containing SNN-unfriendly operators (*e.g.*, softmax), leading to accuracy degradation during SNN fine-tuning. More details of challenges are specified in Section 3.

To address the challenges in HT algorithm, we propose UNISPIKE, an HT framework that combines the high accuracy of A2S with the low latency of DT. To deal with challenges brought by the activation-neuron mismatch and the SNN-unfriendly operators, UNISPIKE proposes **Unified Clip**. Unified Clip is equivalent to ST-BIF neuron and provides a unified replacement for softmax, layer normalization, and GeLU in ANNs. Based on Unified Clip, UNISPIKE further constructs a novel vision transformer, called **UniFormer**, which uses addition-only computation, supports step-by-step inference, and removes *softmax*, *layer normalization*, and *GeLU*. UniFormer performs well in both ANN pre-training and SNN fine-tuning. The contributions of UNISPIKE are summarized as:

▷ We propose the **Unified Clip** operator, which is equivalent to the ST-BIF neuron and can replace *softmax*, *layer normalization*, and *GeLU* without degrading ANN accuracy.

▷ Based on Unified Clip, we propose **UniFormer**, a novel transformer where matrix multiplications can be implemented with addition-only operations, enabling step-by-step inference and performing well in both ANN pre-training and SNN fine-tuning.

▷ UNISPIKE achieves 80.86% accuracy on ImageNet-1K, outperforming the SOTA additional-only and step-by-step DT algorithm Spike-Driven TransformerV2 (80.00%) with 4 time-steps. Compared to the SOTA A2S algorithm SpikeZIP-TF (You et al., 2024), UNISPIKE reduces $5.7\times$ energy with comparable accuracy. To the best of our knowledge, this is the first work to validate HT algorithms on large-scale datasets (*e.g.* ImageNet) using a transformer backbone.

## 2 BACKGROUND AND RELATED WORK

### 2.1 SPIKING NEURON

A spiking neuron is the basic component of SNN, integrating inputs modulated by synaptic weights and outputting spike trains. To overcome the low accuracy issue in A2S algorithm, a spiking neuron called *bipolar integrate&fire with spike tracer* (*aka.* ST-BIF) is proposed (You et al., 2024). For $i$-th pre-synaptic ST-BIF neuron, its output at time-step[1] $t$ is $x_{i,t}^{\text{in}} \in \{-1, 0, 1\}$, and its connection strength (*i.e.*, synaptic weight) with post-synaptic neuron is $w_i$. Each ST-BIF neuron contains a memory unit called *membrane* $V_t$, which can be modeled as a capacitor with a capacitance of $C$. The calculation procedure of ST-BIF neuron is modeled below:

$$\hat{V}_t = V_{t-1} + (\sum_i w_i \cdot x_{i,t})/C \tag{1}$$

$$\Theta(\hat{V}_t, V_{\text{thr}}, S_{t-1}) = \begin{cases} 1; & \hat{V}_t \geq V_{\text{thr}} \ \& \ S_{t-1} < S_{\max} \\ 0; & \text{other} \\ -1; & \hat{V}_t < 0 \ \& \ S_{t-1} > S_{\min} \end{cases}, \quad y_t = \Theta(\hat{V}_t, V_{\text{thr}}, S_{t-1}) \times q_{\text{thr}} \tag{2}$$

$$V_t = \hat{V}_t - y_t/C; \quad S_t = S_{t-1} + \Theta(\hat{V}_t, V_{\text{thr}}, S_{t-1}) \tag{3}$$

where $\hat{V}_t$ is the membrane potential after integration, $y_t$ is the output spikes at $t$ time-step and $V_{\text{thr}}$ is the firing threshold. $S_t$ is an additional memory unit in ST-BIF neuron (*aka.*, spike tracer), and $S_{\max}, S_{\min}$ are the maximum and minimum values of the spike tracer. Equations (1) to (3) describe the dynamics of spike integration, neuron firing, and membrane updating, respectively. To facilitate gradient derivation of the decision function, we rewrite Equation (2) as follows:

$$\Theta(\hat{V}_t, V_{\text{thr}}, S_{t-1}) = \theta(\hat{V}_t - V_{\text{thr}}) \cdot \theta(S_{\max} - S_{t-1} - \varepsilon) - \theta(-\hat{V}_t - \varepsilon) \cdot \theta(S_{t-1} - S_{\min} - \varepsilon) \tag{4}$$

where $\theta$ is the step function. We use $\varepsilon$ to express $>$ and $<$ comparison in Equation (2).

### 2.2 SNN LEARNING ALGORITHM

**ANN-to-SNN (A2S) Conversion Algorithm** *transfers the parameters of the pre-trained ANN into its SNN counterpart* that yields close-to-ANN accuracy (Malcolm & Casco-Rodriguez, 2023). As shown in Table 1, previous A2S algorithms such as MST (Wang et al., 2023) and QCFS (Bu et al., 2023) use an integrate-and-fire (IF) neuron to replace the quantized ReLU function. In recent studies, to reduce the conversion loss caused by the non-equivalence between IF neuron and the quantized ReLU

---

[1]time-step: also called step, a suitable time interval whose length is adequate for once update in the system, including all spikes in the synapse to transport, all the neurons to integrate, and fire spikes once.

Table 1: **Summary of Learning Algorithm.** A2S: ANN-to-SNN conversion. DT: directly training. HT: hybrid training. QANN Eq.: if equivalence to QANN. #SOP: the number of synaptic operations.

| Work | Type | Structure | Neuron | Surrogate | Time-step | #SOP | QANN Eq. | Unfriendly Op. |
|------|------|-----------|--------|-----------|-----------|------|----------|----------------|
| SpikformerV2 (Zhou et al., 2024b) | DT | Transformer | LIF | Sigmoid | 4 | Low | ✗ | SEW-Shortcut |
| SDformerV2 (Yao et al., 2024a) | DT | Transformer | LIF | Sigmoid | 4 | Low | ✗ | SEW-Shortcut |
| QKFormer (Zhou et al., 2024a) | DT | Transformer | LIF | Sigmoid | 4 | Low | ✗ | SEW-Shortcut |
| E-SpikeFormer (Yao et al., 2025) | DT | Transformer | IF | n/a | 8 | Low | ✗ | Temporal Addition |
| QCFS (Bu et al., 2023) | A2S | CNN | IF | n/a | 256 | High | ✗ | LayerNorm |
| MST (Wang et al., 2023) | A2S | Transformer | IF | n/a | 512 | High | ✗ | LayerNorm |
| SpikeZIP-TF (You et al., 2024) | A2S | Transformer | ST-BIF | n/a | 64 | High | ✔ | LayerNorm |
| HYB-SNN (Abuhajar et al., 2025) | HT | CNN | IF | BP | 5 | Low | ✗ | None |
| JAS-SNN (Baltes et al., 2023) | HT | CNN | LIF | Sigmoid | 250 | High | ✗ | None |
| UNISPIKE (Ours) | HT | Transformer | ST-BIF | Normal Distri | 4 | Low | ✔ | None |

Figure 1: **UNISPIKE Pipeline.** UNISPIKE obtained the high-performance SNN by two stages: conversion-stage and learning-stage. (a). In the conversion stage, we build and train an ANN and quantize the activation of the ANN by quantization-aware training. After quantization, we convert the QANN to SNN through layer calibration and neuron replacement. (b) In the learning stage, we train the converted SNN using by BPTT algorithm to obtain a high-performance SNN.

function, ST-BIF neuron is proposed (Hu et al., 2023). However, compared to the direct training (DT) algorithm, the high accuracy of converted SNN requires large inference time-steps ($\geq 64$ time-steps in Table 1), incurring large synaptic operations (*aka.* SOP)[2] and inference latency.

**Direct Learning (DT) Algorithm** *trains the SNN with the back-propagation-through-time (aka. BPTT) algorithm at small time-steps (e.g.* 1 or 4 time-steps (Yao et al., 2024a; Zhou et al., 2024b) *in Table 1).* In DT algorithm, leaky IF neuron (*aka.* LIF) is widely adopted (Yao et al., 2024b;a). However, due to the inaccurate gradient estimated by the surrogate function, the accuracy of SNN is lower than that of ANN with the same number of parameters (Yao et al., 2024a; Zhou et al., 2024b).

## 3 CHALLENGES OF HYBRID TRAINING

The hybrid training (HT) algorithm (Rathi & Roy, 2020; Baltes et al., 2023; Abuhajar et al., 2025) combines the advantages of A2S and DT, to achieve high accuracy with ultra-low time-steps. Figure 1 illustrates the HT pipeline in UNISPIKE. To the best of our knowledge, previous works fail to apply HT to advanced models (*e.g.*, Transformers) on large-scale datasets like ImageNet, mainly due to the critical challenges listed as follow:

**Challenge-1: Low accuracy with addition-only and step-by-step operators.** Addition-only computation is critical to energy-efficient neuromorphic hardware (Narayanan et al., 2020; Davies et al., 2018; Akopyan et al., 2015). As shown in Figure 2(a), a 32-bit addition consumes $32\times$ less energy than a 32-bit MAC, which highlights its energy-saving potential. However, addition-only operators require binary or ternary inputs, making accuracy-enhancing designs such as SEW-shortcut in QKFormer (Zhou et al., 2024a) and Spikformer V2 (Zhou et al., 2024b) infeasible. Meanwhile, in terms of step-by-step inference, as shown in Figure 2(b),(c), it processes each time-step independently rather than processing all time-steps simultaneously in layer-by-layer fashion. *This enables early stopping and fast response.* However, it blocks operators such as temporal addition in E-Spikformer (Yao et al., 2025), which aggregate information across time-steps to improve accuracy.

**Challenge-2: Sub-optimal weight initialization caused by activation–neuron mismatch.** Previous HT algorithms (Rathi & Roy, 2020; Abuhajar et al., 2025; Baltes et al., 2023) replace quantized ReLU with LIF or IF neurons during conversion. These neurons are not mathematically equivalent to quantized ReLU under limited time-steps (Bu et al., 2023). This activation–neuron mismatch yields sub-optimal SNN weight initialization and impairs fine-tuning performance.

---

[2]One synaptic operation refers to the operation that the spiking neuron integrates one pre-synaptic spike into the membrane or fires one spike to the post-synaptic neurons.

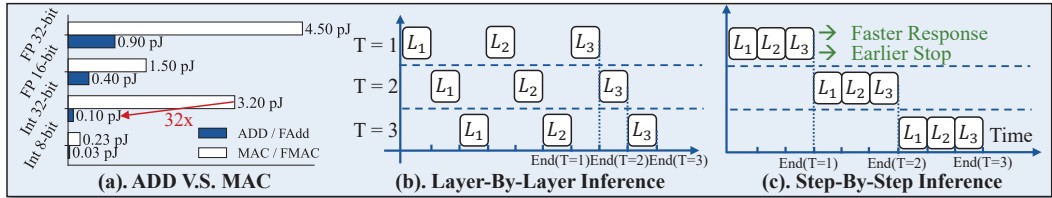

Figure 2: **Advantages of Step-by-step Inference and Addition-Only.** $L_1$ is the 1st SNN layer. (a) Addition consumes less energy than MAC (Horowitz, 2014), showing the energy-saving potential of addition-only SNN. (b) Layer-by-layer inference produces outputs at the first time-step only after all preceding layers have completed their processing across every time-step. (c) Step-by-step inference processes each time-step separately, providing faster response and earlier stopping.

**Challenge-3: Low SNN accuracy after fine-tuning with ANN-specific network structures.** In HT algorithm, ANN and SNN share identical network architecture. However, most modern network architectures, *e.g.*, Vision Transformer (Dosovitskiy, 2020) and ResNet (He et al., 2015), are tailored for ANN trained with back-propagation. Once converting to SNN, these structures may lead to *unstable convergence during SNN fine-tuning* (BPTT) with degraded accuracy.

## 4 METHODOLOGY

To address these challenges, we propose **Unified-Clip**, *a unified activation function that replaces SNN-unfriendly operators in transformers, including layer normalization, softmax, and GeLU*. Unified-Clip preserves addition-only computation and step-by-step inference, and its quantized version can be converted to an equivalent ST-BIF neuron. Based on Unified-Clip, UNISPIKE builds **UniFormer**, a *unified transformer architecture effective for both ANN and SNN training*. Ultimately, to support the SNN fine-tuning, we derive the gradient propagation for ST-BIF neuron.

### 4.1 UNIFIED-CLIP

**Definition.** Unified-Clip in UNISPIKE replaces SNN-unfriendly operators (softmax, layernorm, GeLU), making them support step-by-step inference (challenge-1). Therefore, Unified-Clip must satisfy two key constraints: *1)* Neuron updates in Unified-Clip must be independent across time-steps. Therefore, operators involving data collection from future time-steps are not allowed. *2)* Its quantized form must be mathematically equivalent to ST-BIF neurons. To meet these constraints, we design Unified-Clip ($\text{UC}(x)$) as follows:

$$\text{UC}(x) = \text{clip}(\alpha \cdot x + \beta, X_{\min}, X_{\max}) \cdot \gamma \tag{5}$$

where $x$ is the element of the input tensor $x \in \mathbf{X}$, clip function truncates the input $x$ within $[X_{\min}, X_{\max}]$ and $\alpha, \beta, \gamma$ are the parameters of Unified-Clip. Unified-Clip is an element-wise operation, where the neurons update independently.

**Equivalent to ST-BIF neuron.** We show that quantized Unified-Clip equals the accumulated output of the ST-BIF neuron by leveraging its relation to the quantized function proven in (You et al., 2024):

$$\sum_t^T \text{ST\_BIF}(x_t) = \text{Quan}(\sum_t^T x_t) = \text{clip}(R(\tfrac{\sum_t^T x_t}{s}), C_{\min}, C_{\max}) \cdot s \tag{6}$$

where $x_t$ is the input spikes, $T$ is the total time-steps, $R$ is the rounding function that maps the input to the nearest integer, $s$ is the quantization scale, and $C_{\min}, C_{\max} \in \mathbb{Z}$ are the integer lower and upper bounds. Then, we quantize Unified-Clip, combining Equation (5) and Equation (6) and finally get:

$$\text{Quan}(\text{UC}(\sum_t^T x_t)) = \text{clip}(\text{clip}(R((\alpha \sum_t^T x_t + \beta)\tfrac{\gamma}{s}), R(X_{\min}\tfrac{\gamma}{s}), R(X_{\max}\tfrac{\gamma}{s})), C_{\min}, C_{\max}) \cdot s \tag{7}$$

If we limit $R(X_{\min} \cdot \tfrac{\gamma}{s}) \leq C_{\min}$ and $R(X_{\max} \cdot \tfrac{\gamma}{s}) \geq C_{\max}$, the Equation (7) can be simplfied:

$$\text{Quan}(\text{UC}(\sum_t^T x_t)) = \text{clip}(R(\tfrac{\sum_t^T (\alpha x_t \cdot \gamma + \beta \cdot \gamma/T)}{s}), C_{\min}, C_{\max}) \cdot s \tag{8}$$

By comparing Equation (6) and Equation (8), we build the equivalence between the output quantized Unified-Clip and the accumulated output of ST-BIF neuron:

$$\sum_t^T \text{ST\_BIF}(\alpha x_t \cdot \gamma + \beta \cdot \gamma/T) = \text{Quan}(\text{UC}(\sum_t^T x_t)); \quad \text{s.t.} \quad s \leq \min(\tfrac{|X_{\max}| \cdot \gamma}{|C_{\max}|}, \tfrac{|X_{\min}| \cdot \gamma}{|C_{\min}|}) \tag{9}$$

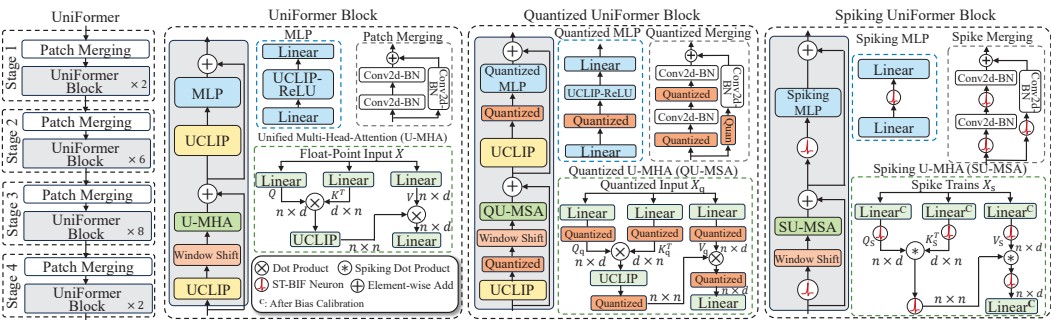

Figure 3: **UniFormer Architecture.** UniFormer consists of 4 stages. Each stage contains a patch merging module and several UniFormer blocks. UniFormer is free of softmax and layer normalization operators and ensures that the inputs of all matrix multiplications are spikes. Therefore, UniFormer is addition-only and supports step-by-step inference.

Equation (9) shows that by limiting the range of quantization scale $s$, quantized Unified Clip is equivalent to ST-BIF neuron. Therefore, by replacing Unified Clip with ST-BIF neuron, the quantized UniFormer is equivalent to its spike version. The equivalence provides better weight initialization for the SNN converted by the A2S compared to replacing quantized ReLU with IF neuron (Challenge-2).

## 4.2 UNIFORMER

To deal with challenge-3, we create a novel vision transformer with Unified-Clip called UniFormer. As shown in Figure 3, UniFormer leverages Unified-Clip to eliminate operators (*e.g.*, softmax), which break the step-by-step feature in SNN. To enhance the performance of UniFormer in the vision task, we use the residual convolution block as the patch merging module at the beginning of all stages.

**UniFormer Block.** A UniFormer block consists of a unified multi-head attention layer, an MLP layer, a window shift module (Liu et al., 2021), and two Unified-Clip modules as a replacement of layer normalization. Specifically, we use Unified-Clip with different $\alpha, \beta, \gamma, C_{\min}, C_{\max}$ to substitute the softmax, layer normalization and GeLU modules in traditional transformers.

**Softmax Replacement.** Softmax normalizes input tensors between 0 and 1, avoiding extremely high activation while maintaining the activation distribution (Wortsman et al., 2023). However, in SNN, spikes from all time steps are necessary for softmax calculation, which is inapplicable to step-by-step processing. Therefore, in UniFormer, we use Unified-Clip to substitute softmax:

$$\text{UC}_{\text{softmax}}(x) = \text{clip}(1/N \cdot x, 0, 1) \quad (10)$$

where $N$ is the number of tokens. We set $C_{\min} = 0, C_{\max} = 1$ to clamp the input to [0, 1] as softmax does. Then, we set $\alpha = 1/N, \beta = 0, \gamma = 1$ to preserve the input distribution as closely as possible.

**Layer Normalization Replacement.** Layer normalization normalizes across the feature dimension of each individual sample. It reduces fluctuations in the input distribution to each layer during training, making training more stable (Ba et al., 2016). However, similar to softmax, layer normalization also uses spikes from all time steps for normalization in SNN, violating the step-by-step inference. (Zhu et al., 2025) finds that the input-output mapping of layer normalization can be approximated with dynamic tanh. However, dynamic tanh is not equivalent to ST-BIF neuron. Therefore, UniFormer uses Unified-Clip to approximate layer normalization by using hard-tanh:

$$\text{UC}_{\text{LayerNorm}}(x) = \text{clip}(\alpha \cdot x + \beta, -1, 1) \cdot \gamma \quad (11)$$

where $C_{\min} = -1$, $C_{\max} = 1$, and $\alpha, \beta, \gamma$ are learnable parameters. Unified-Clip is similar to dynamic hard-tanh (Zhu et al., 2025), but differs in the position of the affine transformation. Unified-Clip applies the affine transform before clipping, rather than after, making the equivalence constraint in Equation (9) easier to satisfy.

**GeLU Replacement.** GeLU activation function (Lee, 2023) performs well in vision transformer. However, GeLU function is not equivalent to ST-BIF neuron. Therefore, we replace GeLU with the

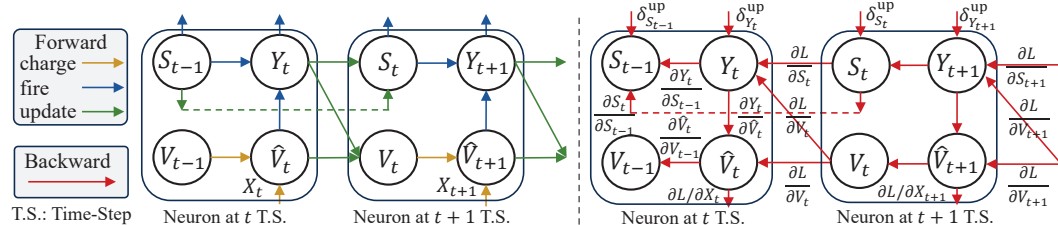

Figure 4: **Computation Graph of ST-BIF Neuron.** In the forward pass, ST-BIF neuron charges its membrane $V_{t-1}$ with input $X_t$, fires spikes and updates its membrane potential $V_{t-1}$ and spike tracer $S_{t-1}$. In the backward pass, ST-BIF neuron receives gradients from the same neuron at the next time-step and neurons in the next layer. Then, it calculates the gradient for weight update.

ReLU function and use Unified-Clip to express ReLU:

$$\text{UC}_{\text{ReLU}}(x) = \text{clip}(x, 0, \infty) \tag{12}$$

where we set $\alpha = 1, \beta = 0, \gamma = 1, C_{\min} = 0, C_{\max} = \infty$. By replacing GeLU with Unified-Clip, we ensure the equivalence between ANN and SNN while maintaining the ANN performance.

### 4.3 BACKWARD PROPAGATION THROUGH TIME (BPTT) FOR ST-BIF NEURON

**Backward Gradient of ST-BIF neuron.** Figure 4 shows the computation graph of ST-BIF neuron. After forwarding, we calculate the gradients through the opposite direction of the forward computing graph. As displayed in the backward pass in Figure 4 (right), ST-BIF neuron at $t$ time-step receives partial derivatives $\delta^{\text{up}}_{S_{t-1}}, \delta^{\text{up}}_{Y_t}$ from neurons in the next layer. Then, it receives partial derivatives $\partial\mathcal{L}/\partial V_t, \partial\mathcal{L}/\partial S_t$ from the same neuron at next time-step $t+1$. After receiving these partial derivatives, the neuron calculates the partial derivatives to the same neuron at the last time-step and neurons in last layers, including $\partial\mathcal{L}/\partial V_{t-1}, \partial\mathcal{L}/\partial S_{t-1}$ and $\partial\mathcal{L}/\partial X_t$:

$$\partial\mathcal{L}/\partial V_{t-1} = (\partial\mathcal{L}/\partial Y_t + \partial\mathcal{L}/\partial S_t - \partial\mathcal{L}/\partial V_t)(\partial\Theta(\hat{V}_t, V_{\text{thr}}, S_t)/\partial\hat{V}_t) + \partial\mathcal{L}/\partial V_t$$

$$\partial\mathcal{L}/\partial S_{t-1} = (\partial\mathcal{L}/\partial Y_t - \partial\mathcal{L}/\partial V_t)(\partial\Theta(\hat{V}_t, V_{\text{thr}}, S_t)/\partial S_{t-1}) + \partial\mathcal{L}/\partial S_t \tag{13}$$

$$\partial\mathcal{L}/\partial X_t = (\partial\mathcal{L}/\partial Y_t + \partial\mathcal{L}/\partial S_t - \partial\mathcal{L}/\partial V_t)(\partial\Theta(\hat{V}_t, V_{\text{thr}}, S_t)/\partial\hat{V}_t) + \partial\mathcal{L}/\partial V_t$$

The detailed derivation is in the appendix (Section A1). $\partial\Theta(\hat{V}_t, V_{\text{thr}}, S_t)/\partial\hat{V}_t, \partial\Theta(\hat{V}_t, V_{\text{thr}}, S_t)/\partial S_{t-1}$ are partial derivatives of fire decision function, which is non-differentiable. Therefore, we use a surrogate gradient to estimate the gradients.

**Surrogate Gradient of ST-BIF neuron.** In ST-BIF neuron, the fire decision function $\Theta$ is expressed by several step functions in Equation (4). Therefore, we expand the partial derivatives of $\Theta$ as follows:

$$\partial\Theta(\cdot)/\partial\hat{V}_t = \theta'(\hat{V}_t - V_{\text{thr}}) \cdot \theta(S_{\max} - S_{t-1} - \varepsilon) + \theta'(-\hat{V}_t - \varepsilon) \cdot \theta(S_{t-1} - S_{\min} - \varepsilon)$$

$$\partial\Theta(\cdot)/\partial S_{t-1} = -\theta(\hat{V}_t - V_{\text{thr}}) \cdot \theta'(S_{\max} - S_{t-1} - \varepsilon) - \theta(-\hat{V}_t - \varepsilon) \cdot \theta'(S_{t-1} - S_{\min} - \varepsilon) \tag{14}$$

where $\theta'$ is Dirac Delta function. Since Dirac Delta function is hard to implement on the GPU platform, we use the sigmoid function to estimate $\theta$:

$$\sigma(x) = 1/(1 + e^{-\mu x}) \tag{15}$$

where $\mu$ is a constant, controlling the steepness of the sigmoid function.

## 5 EXPERIMENT

**Experiment Setup.** Various vision datasets are adopted for evaluation. **1) static vision datasets**, including CIFAR10 (Krizhevsky et al., 2009), CIFAR100 (Krizhevsky et al., 2009) and ImageNet (Deng et al., 2009). **2) neuromorphic vision dataset**: We evaluate UNISPIKE on CIFAR10-DVS (Hongmin et al., 2017). On ImageNet, we apply the standard UniFormer architecture (66.5M) with 128 input embedding dimension and the four stages contains [2,6,8,2] UniFormer blocks, respectively. For CIFAR-10/100 and CIFAR10-DVS, we utilize the UniFormer architecture with a smaller embedding

Table 2: **Comparison between SOTA algorithms on ImageNet.** $\triangle$ indicates that non-spike operations are treated as MAC operations in terms of their energy consumption. † refers to the QANN without weight quantization. The best results are in **bold** and runner-up results are in gray with addition-only and step-by-step inference with comparable energy consumption.

| Methods | Cate. | Architecture | Addition Only | Step-by-Step | Param (M) | Power (mJ) | Step | Acc.(%) |
|---|---|---|---|---|---|---|---|---|
| ANN | BP | RegNetY-16G | ✗ | ✗ | 84.0 | - | 1 | 82.90 |
| | | ViT-B | ✗ | ✗ | 86.0 | - | 1 | 82.30 |
| | | Swin-B | ✗ | ✗ | 88.0 | - | 1 | **83.50** |
| | | T2T-ViT | ✗ | ✗ | 64.3 | - | 1 | 82.30 |
| | | UniFormer | ✗ | ✗ | 65.5 | - | 1 | 83.30 |
| QANN† | BP | UniFormer | ✗ | ✗ | 66.5 | 62.30 | 1 | 81.63 |
| DieT-SNN | HT | VGG16 | ✔ | ✔ | 138.4 | - | 5 | 69.00 |
| Hybrid-STDP | HT | VGG16 | ✔ | ✔ | 138.4 | - | 250 | 65.19 |
| Fast-SNN | A2S | VGG16 | ✔ | ✔ | 138.4 | - | 7 | 72.95 |
| MST | A2S | Swin-T(BN) | ✔ | ✔ | 28.5 | - | 512 | 78.51 |
| SpikeZip-TF | A2S | SViT-S-32Level | ✔ | ✔ | 22.05 | 102.7 | 64 | 81.45 |
| | | SViT-B-32Level | ✔ | ✔ | 86.57 | 403.2 | 64 | 82.71 |
| SpikingFormer | DT | Spikingformer-8-512 | ✔ | ✔ | 29.68 | 7.46 | 4 | 74.79 |
| | | Spikingformer-8-768 | ✔ | ✔ | 66.34 | 13.68 | 4 | 75.85 |
| E-SpikeFormer | DT | E-SpikeFormer | ✔ | ✗ | 19.0 | 5.9 | 4 | 79.80 |
| | | | ✔ | ✗ | 83.0 | 19.1 | 4 | 83.20 |
| QKFormer | DT | HST-10-384 | ✗ | ✔ | 29.08 | 21.99/104.04$^\triangle$ | 4 | 82.04 |
| | | HST-10-768 | ✗ | ✔ | 64.96 | 38.91/231.89$^\triangle$ | 4 | 84.22 |
| SDFormerV2 | DT | Meta-SpikeFormer | ✔ | ✔ | 15.1 | 16.7 | 4 | 74.10 |
| | | | ✔ | ✔ | 31.3 | 32.8 | 4 | 77.20 |
| | | | ✔ | ✔ | 55.4 | 52.4 | 4 | 80.00 |
| UniSpike | HT | UniFormer | ✔ | ✔ | 66.5 | 18.0 | 4 | **80.83** |

Table 3: **Experimental results on CIFAR-10, CIFAR-100 and CIFAR10-DVS.** *CF* is the abbreviation of CIFAR. The best results are in **bold**, the runner-up results are in gray.

| Methods | Category | Arch. | Param. | T | CF10 | CF100 | Arch. | Param. | T | CF10-DVS |
|---|---|---|---|---|---|---|---|---|---|---|
| ResNet-19 | ANN | Conv. | 12.63 | 1 | 94.97 | 75.35 | n/a | n/a | n/a | n/a |
| Transformer | ANN | 4-384 | 9.32 | 1 | 96.73 | 81.02 | n/a | n/a | n/a | n/a |
| Spikformer | SNN | 4-384 | 9.32 | 4 | 95.19 | 77.86 | 2-256 | 2.57 | 16 | 80.9 |
| Spikingformer | SNN | 4-384 | 9.32 | 4 | 95.81 | 78.21 | 2-256 | 2.57 | 16 | 81.3 |
| CML | SNN | 4-384 | 9.32 | 4 | 96.04 | 80.02 | 2-256 | 2.57 | 16 | 80.9 |
| S-Transformer | SNN | 4-384 | 10.28 | 4 | 95.60 | 78.40 | 2-256 | 2.57 | 16 | 80.0 |
| QKFormer | SNN | 4-384 | 6.74 | 4 | 96.18 | 81.15 | 2-256 | 1.50 | 16 | 84.0 |
| UniSpike | SNN | UniFormer-T1 | 4.50 | 6 | **97.16** | **82.73** | UniFormer-T2 | 2.15 | 8 | **85.0** |

dimension (*aka.* UniFormer-T1, UniFormer-T2) to make a fair comparison with other SNN works. The network details are in the appendix (Table A5). In quantization-aware training (*aka.* QAT), we conduct activation quantization with 10 quantization levels. To satisfy the equivalence condition in Equation (9), we limit the quantization scale $s$ less than $\min(\frac{|X_{max}| \cdot \gamma}{|C_{max}|}, \frac{|X_{min}| \cdot \gamma}{|C_{min}|})$ during QAT. Note that we introduce distillation on both the ANN training and the QAT procedure. The teacher model is Swin-B (Liu et al., 2021) pretrained on ImageNet-21K. For the sigmoid function in Equation (15), UNISPIKE set $\mu = 4$.

## 5.1 RESULTS ON IMAGENET-1K CLASSIFICATION

**Comparison with ANN Architectures.** As shown in Table 2, we compare UniFormer with classic ANN architecture including ViT-B (Dosovitskiy, 2020), Swin-B (Liu et al., 2021), RegNet-Y (Radosavovic et al., 2020) and T2T-ViT (Yuan et al., 2021). UniFormer achieves the second-best accuracy (83.3%), less than the Swin-B (83.5%) with fewer parameters (67M v.s. 88M). Notably, UniFormer replaces softmax, layer normalization, and GeLU activation with unified-clip, indicating that Unified-Clip serves as an effective substitute during ANN training.

**Comparison with SOTA SNN Algorithms.** We compare UNISPIKE with the previous competitive SNN works, including SpikFormerV2 (Zhou et al., 2024b), spike-driven transformer V2 (*aka.* SDFromerV2) (Yao et al., 2024a), SpikeZIP-TF (You et al., 2024) and QKFormer (Zhou et al., 2024a) in Table 2. Although E-Spikformer (Yao et al., 2025) has higher accuracy, it does not support

step-by-step inference, and its energy consumption simulation with spike-driven inference is highly dependent on the specific neuromorphic hardware (Man et al., 2024). **Compared to other DT algorithm**, QKFormer (Zhou et al., 2024a) achieves the highest accuracy. However, QKFormer contains a large amount of MAC operations as shown in Table 4, causing 12.9 × energy consumption compared to UNISPIKE with comparable parameters (HST-10-384 V.S. UniFormer). Compared to other addition-only SNNs with DT algorithm, thanks to the high accuracy provided by the conversion algorithm, UNISPIKE achieves the best accuracy, surpassing Spike-Driven transformer V2 with comparable parameters. **Compared to the SOTA A2S algorithm**, SpikeZIP-TF (You et al., 2024), since UNISPIKE uses BPTT algorithm to compress the inference time-step, UNISPIKE achieves 5.7× energy reduction and 16× time-step reduction with comparable accuracy (80.85% in UniFormer V.S. 81.45% SViT-S-32Level). **Compared to the QANN**, thanks to the addition-only feature, UNISPIKE achieves QANN comparable accuracy with less energy consumption.

## 5.2 RESULTS ON CIFAR AND NEUROMORPHIC DATASETS

On CIFAR and neuromorphic datasets, UNISPIKE achieves the SOTA performance. UNISPIKE achieves 0.98% and 1.58% higher task accuracy on CIFAR10 and CIFAR100 compared to QKFormer with smaller parameters (4.50M V.S. 6.74M in QKFormer) at 6 time-step. On the temporal neuromorphic dataset CIFAR10-DVS, UNISPIKE SNN has 0.2% higher accuracy compared to QKFormer with fewer time-steps (8 time-steps v.s. 16 time-steps in QKFormer), but with larger parameters (2.15 M v.s. 1.50 M in QKFormer).

## 5.3 OPERATION & ENERGY ANALYSIS

To further demonstrate the benefit of addition-only SNN, we analyze the # of ACs and MACs breakdown of SpikingFormer-8-768 (Zhou et al., 2023), QKFormer-HST-10-384 (Zhou et al., 2024a) and UniFormer, excepting the patch embedding layer and head layer, which are shown in Table 4. To calculate energy consumption, following previous works (Yao et al., 2024b;a; Zhou et al., 2023), we use 0.9 pJ/AC and 4.5 pJ/MAC data from (Horowitz, 2014). In Table 4, we have three observations: 1) In QK-Former, MAC operation consumes more energy than AC operation, causing the energy inefficiency. Meanwhile, since the UniFormer and SpikingFormer are addition-only SNNs, all the operations are AC operations, achieving substantial energy reduction. 2) Since the SEW shortcut introduces non-spiking input, the MAC opera-

Table 4: **Operation of Transformer-Based SNNs.** $Q, K, V$ are the layers that generate query, key, and value in attention. $M$ is the multiplication between query and key or the multiplication between key and value. $A$ is attention multiplication. $Y$ is the projection layer in attention.

| Transformer Layers | | SpikingFo. | QKFormer | | UniFormer |
|---|---|---|---|---|---|
| | | GACs | GACs | GMACs | GACs |
| | $Q$ | 0.53 | 0.00 | 2.06 | 1.89 |
| | $K$ | 0.53 | 0.00 | 2.06 | 1.89 |
| Self-Attention | $V$ | 0.53 | 0.00 | 1.44 | 1.89 |
| | $M$ | 0.001 | 0.01 | 0.00 | 0.99 |
| | $A$ | 0.002 | 0.03 | 0.00 | 0.60 |
| | $Y$ | 0.37 | 0.04 | 0.00 | 0.74 |
| MLP | FC1 | 2.17 | 0.00 | 8.24 | 6.95 |
| | FC2 | 0.12 | 0.24 | 0.00 | 0.49 |
| Patch Embedding | | 2.25 | 2.17 | 0.35 | 1.72 |
| Patch Merging | | 0.00 | 0.85 | 7.60 | 3.30 |
| Total Operation | | **6.52** | 3.63 | 22.09 | 19.97 |
| Energy Cost (mJ) | | **5.87** | 3.23 | 99.40 | 17.97 |

tions in QKFormer appear in the layers after SEW shortcut (Fang et al., 2021), such as $Q, K, V$ and FC1. 3) The number of operation are concentrated on $Q, K, V$ and FC1 layers. The reason is that these layers are after the residual addition, receiving more spikes than other layers.

## 5.4 STEP-BY-STEP INFERENCE ON UNIFORMER

Step-by-step inference provides a practical advantage over layer-by-layer inference by enabling early outputs and reducing response latency. As illustrated in Figure 5, we evaluate UniFormer as the backbone of the Unitrack framework (Wang et al., 2021) on the DAVIS2017 dataset (Pont-Tuset et al., 2017) for video object segmentation. Thanks to its step-by-step inference capability, UniFormer produces an output immediately after completing computations at each time-step. Consequently, the frame rate at $T$=1 (35.5 FPS) is 4.5× faster than that at $T$=4 (7.8 FPS). For simple tracking cases (Image-1), the mask quality at $T$=1 is sufficient, enabling early exiting after the first time-step and achieving inference at 35.5 FPS. For more challenging cases (Image-2), multiple time-steps are required to obtain sufficiently accurate masks, allowing the network to adaptively trade off speed for accuracy by performing inference at a lower FPS. These results demonstrate that step-by-step

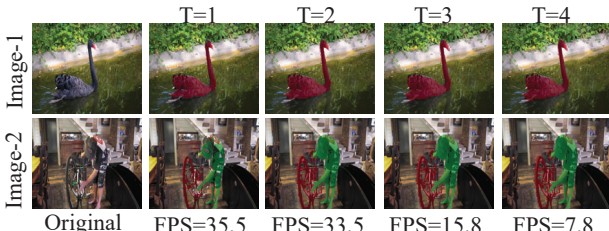
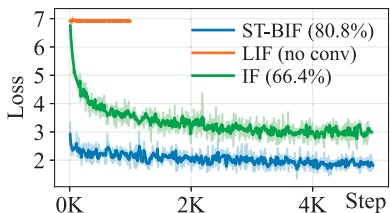

Figure 5: The output of UniFormer at different time-steps on the DAVIS2017 dataset.

Figure 6: Training Loss of UniFormer with different neuron models.

Table 5: Surrogate Function Comparison

| Surrogate Function | CIFAR100 | | CIFAR10-DVS | |
|---|---|---|---|---|
| | QANN | SNN | QANN | SNN |
| Atan | 81.48 | 82.30 | 84.61 | **84.82** |
| SoftSign | 81.48 | 82.32 | 84.61 | 83.92 |
| Sigmoid | 81.48 | **82.74** | 84.61 | 84.42 |

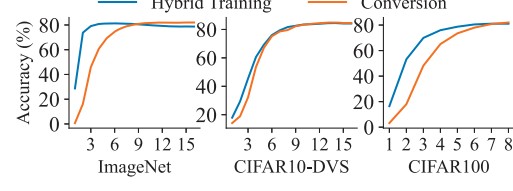

Figure 7: Time-Step versus Accuracy Curve

inference can deliver low-latency outputs when possible while maintaining accuracy for more difficult scenarios, which is crucial for real-time applications.

## 5.5 ABLATION STUDY

**Neuron Type.** The equivalence between Unified-Clip and ST-BIF neuron provides a suitable initialization for SNN fine-tuning. Figure 6 shows the training loss curve of UniFormer during SNN fine-tuning with different neuron models on ImageNet-1K. As depicted, due to the equivalence with Unified-Clip, ST-BIF achieves the lowest training loss. IF neuron converges slowly since it matches the quantization function only as time-steps approach infinity Bu et al. (2023). In contrast, LIF neuron fails to converge because its leaky membrane breaks the equivalence with Unified-Clip.

**Surrogate Function.** To derive the best surrogate function for the decision function $\Theta$ in ST-BIF neuron, we conduct multiple experiments on CIFAR100 and CIFAR10-DVS with different surrogate functions, including atan, softsign and sigmoid. Results in Table 5 demonstrate that sigmoid achieves the best and second-best accuracy on CIFAR100 and CIFAR10-DVS, respectively. Therefore, UNISPIKE uses sigmoid as the surrogate gradient in BPTT.

**Accuracy versus Time-Step** Figure 7 displays the time-step v.s. accuracy curve after conversion and hybrid training on various datasets. After SNN fine-tuning, the SNN accuracies at small time-steps increase significantly (from 16.2% to 73.8% at time-step 2 on ImageNet-1K). However, since the BPTT algorithm optimizes SNN accuracy at a specific time-step, the SNN accuracies at large time-steps decrease (from 82.0% to 78.7% at time-step 16).

## 6 CONCLUSION

In this paper, we propose UNISPIKE, a hybrid learning framework to boost the performance of SNN. UNISPIKE firstly applies a conversion algorithm to ensure the high accuracy of SNN and then conducts BPTT training to improve the accuracy of SNN under ultra-low time-steps for higher energy efficiency. UNISPIKE SNN incorporates an addition-only feature with step-by-step inference, constructing a more energy-efficient architecture than ANN, which is suitable for real-world applications on neuromorphic hardware. We anticipate extending our models to the real-world applications on SNN deployment with neuromorphic hardware, further verifying the effectiveness of UNISPIKE.

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

Table A1: The Notations in Formula Derivation

| Symbol | Description |
|---|---|
| $\mathcal{L}$ | Loss function |
| $w_i$ | synaptic weights connected to $i$-th neuron |
| $X_t$ | Input at $t$ time-step |
| $Y_t$ | Output spikes at $t$ time-step |
| $V_t$ | Membrane potential at $t$ time-step |
| $V_{\mathrm{thr}}$ | Membrane threshold for neuron firing |
| $S_t$ | Spike tracer at $t$ time-step |
| $\hat{V}_t$ | Membrane potential after charging before firing at $t$ time-step |
| $S_{\min}$ | The minimum value of spike tracer |
| $S_{\max}$ | The maximum value of spike tracer |
| $\Theta$ | Fire decision function of ST-BIF Neuron You et al. (2024) |
| $\theta$ | Step function Weisstein (2002) |
| $\varepsilon$ | A small value |

Figure A1: **Computation Graph of ST-BIF Neuron.**

# Appendix

## A1 BPTT FOR SNN WITH ST-BIF NEURON

The forward procedure of ST-BIF neuron is different from LIF neuron. Consequently, deriving the backward gradient of ST-BIF neuron is essential for implementing the BPTT algorithm on the SNN comprised of ST-BIF neurons. For the convenience of derivation, we summarize the meaning of notations appearing in Table A1.

### A1.1 FEED-FORWARD OF ST-BIF NEURON

Before the derivation of backward gradient, we display the forward path of ST-BIF neuron in Figure A1(left). ST-BIF neuron charges the membrane potential $V_{t-1}$ with the input $X_t$ to get $\hat{V}_t$ (yellow arrows in Figure A1), fires spikes according to the firing decision function $\Theta$ (blue arrows in Figure A1) and update the $S_{t-1}, \hat{V}_t$ to the states at the next time-step (green arrows in Figure A1). The feed-forward definition of ST-BIF neuron is below:

$$\hat{V}_t = V_{t-1} + \left(\sum_i w_i \cdot X_{i,t}\right) \tag{A1}$$

$$Y_t = \Theta(\hat{V}_t, V_{\mathrm{thr}}, S_{t-1}) \tag{A2}$$

$$\Theta(\hat{V}_t, V_{\mathrm{thr}}, S_{t-1}) = \theta(\hat{V}_t - V_{\mathrm{thr}}) \cdot \theta(S_{\max} - S_{t-1} - \varepsilon) - \theta(-\hat{V}_t - \varepsilon) \cdot \theta(S_{t-1} - S_{\min} - \varepsilon)$$

$$V_t = \hat{V}_t - V_{\mathrm{thr}} \times Y_t; \quad S_t = S_{t-1} + \Theta(\hat{V}_t, V_{\mathrm{thr}}, S_{t-1}) \tag{A3}$$

where Equation (A1),Equation (A2),Equation (A3) are the charging, firing and updating procedures in ST-BIF neuron, respectively.

### A1.2 BACKWARD OF ST-BIF NEURON

After forwarding, we calculate the gradients through the opposite direction of the forward computing graph. As displayed in the backward pass in Figure A1 (right), ST-BIF neuron at $t$ time-step receives

partial derivatives of the loss function $\mathcal{L}$ to membrane potential $V_t$ and spike tracer $S_t$ from the same neuron at $t+1$ time-step, *i.e.*, $\partial\mathcal{L}/\partial V_t, \partial\mathcal{L}/\partial S_t$, as well as the partial derivatives of $\partial\mathcal{L}/\partial Y_t$ from neurons in next layer (red arrows outside the neuron at $t$ time-step in Figure A1).

After receiving these partial derivatives, the neuron calculates the partial derivatives of the loss function $\mathcal{L}$ to membrane potential $V_{t-1}$, spike tracer $S_{t-1}$ and input $X_t$, *i.e.*, $\partial\mathcal{L}/\partial V_{t-1}, \partial\mathcal{L}/\partial S_{t-1}$ and $\partial\mathcal{L}/\partial X_t$ for the $t-1$ time-step. Then, the neuron transmits $\partial\mathcal{L}/\partial V_{t-1}$ and $\partial\mathcal{L}/\partial S_{t-1}$ to the same neuron at $t-1$ time-step, and sends $\partial\mathcal{L}/\partial X_t$ to the neurons in the last layer. In summary, we need to derive three partial derivatives: $\partial\mathcal{L}/\partial V_{t-1}, \partial\mathcal{L}/\partial S_{t-1}$ and $\partial\mathcal{L}/\partial X_t$.

According to the computation graph in Figure A1 and chain rule, we write the calculation formulation for the three partial derivatives:

$$\frac{\partial\mathcal{L}}{\partial V_{t-1}} = \frac{d\mathcal{L}}{dY_t}\frac{\partial Y_t}{\partial\hat{V}_t}\frac{\partial\hat{V}_t}{\partial V_{t-1}} + \frac{\partial\mathcal{L}}{\partial V_t}\frac{\partial\hat{V}_t}{\partial V_{t-1}}, \quad \frac{\partial\mathcal{L}}{\partial S_{t-1}} = \frac{d\mathcal{L}}{dY_t}\frac{\partial Y_t}{\partial S_{t-1}} + \frac{\partial\mathcal{L}}{\partial S_t}\frac{\partial S_t}{\partial S_{t-1}}$$

$$\frac{\partial\mathcal{L}}{\partial X_t} = \frac{d\mathcal{L}}{dY_t}\frac{\partial Y_t}{\partial\hat{V}_t}\frac{\partial\hat{V}_t}{\partial X_t} + \frac{\partial\mathcal{L}}{\partial V_t}\frac{\partial\hat{V}_t}{\partial X_t}, \quad \frac{d\mathcal{L}}{dY_t} = \frac{\partial\mathcal{L}}{\partial Y_t} + \frac{\partial\mathcal{L}}{\partial S_t}\frac{\partial S_t}{\partial Y_t} + \frac{\partial\mathcal{L}}{\partial V_t}\frac{\partial V_t}{\partial Y_t} \tag{A4}$$

where $\frac{d\mathcal{L}}{dY_t}$ contains all the partial derivatives transmitted to $Y_t$, including $\frac{\partial\mathcal{L}}{\partial Y_t}, \frac{\partial\mathcal{L}}{\partial S_t}$ and $\frac{\partial\mathcal{L}}{\partial V_t}$ (the red arrows to $Y_t$ in Figure A1). Then, combining with the firing decision function of ST-BIF neuron defined in Equation (A2), we calculate the partial derivatives in Equation (A4) and have:

$$\frac{\partial\hat{V}_t}{\partial V_{t-1}} = \frac{\partial S_t}{\partial Y_t} = \frac{\partial\hat{V}_t}{\partial X_t} = 1, \frac{\partial V_t}{\partial Y_t} = -V_{\text{thr}} \tag{A5}$$

$$\frac{\partial S_t}{\partial S_{t-1}} = 1 - \frac{\partial Y_t}{\partial S_{t-1}} \tag{A6}$$

$$\frac{\partial Y_t}{\partial\hat{V}_t} = \frac{\partial\Theta(\hat{V}_t, V_{\text{thr}}, S_t)}{\partial\hat{V}_t}, \quad \frac{\partial Y_t}{\partial S_{t-1}} = \frac{\partial\Theta(\hat{V}_t, V_{\text{thr}}, S_t)}{\partial S_{t-1}} \tag{A7}$$

$$\frac{\Theta(\hat{V}_t, V_{\text{thr}}, S_t)}{\partial\hat{V}_t} = \theta'(\hat{V}_t - V_{\text{thr}})\cdot\theta(S_{\max} - S_{t-1} - \varepsilon) + \theta'(-\hat{V}_t - \varepsilon)\cdot\theta(S_{t-1} - S_{\min} - \varepsilon) \tag{A8}$$

$$\frac{\partial\Theta(\hat{V}_t, V_{\text{thr}}, S_t)}{\partial S_{t-1}} = -\theta(\hat{V}_t - V_{\text{thr}})\cdot\theta'(S_{\max} - S_{t-1} - \varepsilon) - \theta(-\hat{V}_t - \varepsilon)\cdot\theta'(S_{t-1} - S_{\min} - \varepsilon) \tag{A9}$$

where $\theta'$ is Dirac delta function Hassani & Hassani (2009), which is the derivative of $\theta$. Since Dirac delta function is hard to implement in hardware, we use the normal distribution function as the surrogate function of $\theta'$. Then, we substitute Equation (A5)-Equation (A9) to Equation (A4) and finally get:

$$\frac{\partial\mathcal{L}}{\partial V_{t-1}} = (\frac{\partial\mathcal{L}}{\partial Y_t} + \frac{\partial\mathcal{L}}{\partial S_t} - V_{\text{thr}}\times\frac{\partial\mathcal{L}}{\partial V_t})\frac{\partial Y_t}{\partial\hat{V}_t} + \frac{\partial\mathcal{L}}{\partial V_t}$$

$$\frac{\partial\mathcal{L}}{\partial S_{t-1}} = (\frac{\partial\mathcal{L}}{\partial Y_t} + \frac{\partial\mathcal{L}}{\partial S_t} - V_{\text{thr}}\times\frac{\partial\mathcal{L}}{\partial V_t})\frac{\partial Y_t}{\partial S_{t-1}} + \frac{\partial\mathcal{L}}{\partial S_t}(1 - \frac{\partial Y_t}{\partial S_{t-1}}) \tag{A10}$$

$$\frac{\partial\mathcal{L}}{\partial X_t} = (\frac{\partial\mathcal{L}}{\partial Y_t} + \frac{\partial\mathcal{L}}{\partial S_t} - V_{\text{thr}}\times\frac{\partial\mathcal{L}}{\partial V_t})\frac{\partial Y_t}{\partial\hat{V}_t} + \frac{\partial\mathcal{L}}{\partial V_t}$$

where we do not substitute $\frac{\partial Y_t}{\partial\hat{V}_t}$ and $\frac{\partial Y_t}{\partial S_{t-1}}$ for the convenient of reading. After simply transforming, we finally get:

$$\frac{\partial\mathcal{L}}{\partial V_{t-1}} = (\frac{\partial\mathcal{L}}{\partial Y_t} + \frac{\partial\mathcal{L}}{\partial S_t} - V_{\text{thr}}\times\frac{\partial\mathcal{L}}{\partial V_t})\frac{\Theta(\hat{V}_t, V_{\text{thr}}, S_t)}{\partial\hat{V}_t} + \frac{\partial\mathcal{L}}{\partial V_t}$$

$$\frac{\partial\mathcal{L}}{\partial S_{t-1}} = (\frac{\partial\mathcal{L}}{\partial Y_t} - V_{\text{thr}}\times\frac{\partial\mathcal{L}}{\partial V_t})\frac{\partial\Theta(\hat{V}_t, V_{\text{thr}}, S_t)}{\partial S_{t-1}} + \frac{\partial\mathcal{L}}{\partial S_t} \tag{A11}$$

$$\frac{\partial\mathcal{L}}{\partial X_t} = (\frac{\partial\mathcal{L}}{\partial Y_t} + \frac{\partial\mathcal{L}}{\partial S_t} - V_{\text{thr}}\times\frac{\partial\mathcal{L}}{\partial V_t})\frac{\Theta(\hat{V}_t, V_{\text{thr}}, S_t)}{\partial\hat{V}_t} + \frac{\partial\mathcal{L}}{\partial V_t}$$

With Equation (A11), we correctly estimate the gradient for ST-BIF neuron and make the SNN in UNISPIKE support the BPTT training algorithm.

## A2 EXPERIMENT SETUPS

### A2.1 TRAINING SETTINGS

For the reproduction of our experiment results, we show detailed training settings in Tables A2 to A4, including training hyperparameters, data augmentations, and other training technologies.

Table A2: **ANN Pre-training, Quantization-Aware Training and SNN Training Setting on ImageNet (UniFormer).**

| config | ANN | QANN | SNN |
|---|---|---|---|
| optimizer | AdamW | AdamW | AdamW |
| learning rate | 1e-3 | 1e-4 | 2.5e-5 |
| weight decay | 0.05 | 0.05 | 0.0005 |
| optimizer momentum | $\beta_1, \beta_2=0.9, 0.999$ | $\beta_1, \beta_2=0.9, 0.999$ | $\beta_1, \beta_2=0.9, 0.999$ |
| layer-wise lr decay Clark et al. (2020); Bao et al. (2022) | None | None | None |
| batch size | 1024 | 1024 | 128 |
| learning rate schedule | cosine decay | cosine decay | cosine decay |
| warmup epochs | 20 | 5 | 0 |
| training epochs | 300 | 40 | 10 |
| augmentation | RandAug (9, 0.5) Cubuk et al. (2019) | RandAug (9, 0.5) Cubuk et al. (2019) | RandAug (9, 0.5) Cubuk et al. (2019) |
| label smoothing Szegedy et al. (2016) | 0.1 | 0.1 | 0.1 |
| mixup Zhang et al. (2018) | 0.8 | 0.5 | None |
| cutmix Yun et al. (2019) | 1.0 | 1.0 | None |
| drop path Huang et al. (2016) | 0.1 | 0.1 | 0.1 |

Table A3: **ANN pretraining, Quantization-Aware Training and SNN Training Setting on CI-FAR100 (UniFormer-T1).**

| config | ANN | QANN | SNN |
|---|---|---|---|
| optimizer | AdamW | AdamW | AdamW |
| learning rate | 4e-3 | 4e-4 | 5e-4 |
| weight decay | 0.05 | 0.05 | 0.05 |
| optimizer momentum | $\beta_1, \beta_2=0.9, 0.999$ | $\beta_1, \beta_2=0.9, 0.999$ | $\beta_1, \beta_2=0.9, 0.999$ |
| layer-wise lr decay Clark et al. (2020); Bao et al. (2022) | None | None | None |
| batch size | 1024 | 512 | 128 |
| learning rate schedule | cosine decay | cosine decay | cosine decay |
| warmup epochs | 10 | 5 | 5 |
| training epochs | 300 | 100 | 100 |
| augmentation | RandAug (9, 0.5) Cubuk et al. (2019) | RandAug (9, 0.5) Cubuk et al. (2019) | RandAug (9, 0.5) Cubuk et al. (2019) |
| label smoothing Szegedy et al. (2016) | 0.1 | 0.1 | 0.1 |
| mixup Zhang et al. (2018) | 0.5 | 0.5 | None |
| cutmix Yun et al. (2019) | 1.0 | 1.0 | None |
| drop path Huang et al. (2016) | 0.1 | 0.1 | 0.1 |

### A2.2 NETWORK ARCHITECTURE DETAILS

Table A5 displays the architecture details of UNISPIKE, including UniFormer, UniFormer-T1 and UniFormer-T2. All the patch embedding and patch merging blocks in UNISPIKE SNN are convolution

Table A4: **ANN pretraining, Quantization-Aware Training and SNN Training Setting on CIFAR10-DVS (UniFormer-T2).**

| config | ANN | QANN | SNN |
|---|---|---|---|
| optimizer | AdamW | AdamW | AdamW |
| learning rate | 2e-3 | 5e-3 | 5e-5 |
| weight decay | 0.5 | 0.5 | 0.0005 |
| optimizer momentum | $\beta_1, \beta_2{=}0.9, 0.999$ | $\beta_1, \beta_2{=}0.9, 0.999$ | $\beta_1, \beta_2{=}0.9, 0.999$ |
| layer-wise lr decay Clark et al. (2020); Bao et al. (2022) | None | None | None |
| batch size | 512 | 256 | 128 |
| learning rate schedule | cosine decay | cosine decay | cosine decay |
| warmup epochs | 10 | 0 | 0 |
| training epochs | 400 | 200 | 20 |
| augmentation | SNNAugmentWide Müller & Hutter (2021) | SNNAugmentWide Müller & Hutter (2021) | SNNAugmentWide Müller & Hutter (2021) |
| label smoothing Szegedy et al. (2016) | 0.1 | 0.1 | 0.1 |
| mixup Zhang et al. (2018) | 0.5 | 0.5 | 0.5 |
| cutmix Yun et al. (2019) | 1.0 | 1.0 | 1.0 |
| drop path Huang et al. (2016) | 0.1 | 0.1 | 0.1 |

residual blocks in ResNet. The input resolution of all the datasets is 224x224 to fully utilize the pretrained ANN on ImageNet.

Table A5: The Network Architecture Details of UNISPIKE.

| Name | Resolution | Patch Size | Window Size | Embedding Dim | Depths | Num Heads |
|---|---|---|---|---|---|---|
| UniFormer | $224 \times 224$ | 4 | 7 | 128 | [2, 6, 8, 2] | [4, 8, 16, 32] |
| UniFormer-T1 | $224 \times 224$ | 4 | 7 | 36 | [2, 2, 6, 2] | [1, 3, 6, 12] |
| UniFormer-T2 | $224 \times 224$ | 4 | 7 | 24 | [2, 2, 6, 2] | [1, 3, 6, 12] |

## A3   USE OF LLMS

We leverage LLMs to aid or polish writing. Specifically, LLMs help us find some grammar and spelling mistakes after we finish writing.

