# OpenReview forum: "UniSpike: Boosting the Performance of Spiking Neural Network with Hybrid Training"
_ICLR.cc/2026/Conference — Submitted to ICLR 2026_

### Official Review · Reviewer_XRYA · 2025-10-20

**Soundness:** 2
**Presentation:** 3
**Contribution:** 3
**Rating:** 6
**Confidence:** 3

**Summary:**

This paper proposes a unified clip approach for hybrid training of Spiking Neural Networks (SNNs), which transforms non–SNN-friendly operators in Artificial Neural Networks (ANNs) into SNN-compatible forms. Theoretically, it demonstrates that the proposed clip operator is equivalent to a type of neuron in SNNs. Finally, the converted network is fine-tuned using Backpropagation Through Time (BPTT) to achieve a fully converted SNN with low latency and high performance.

**Strengths:**

1. In the field of hybrid training, the authors improved performance from 69%/65.19% to 80.83%, demonstrating the effectiveness of the proposed framework.

2. A simple yet effective clip function is used to uniformly convert multiple operators, and the equivalence between this clip function and existing spiking neurons in SNNs is also theoretically proven.

3. The literature review is up to date, taking into account recent advances in the field.

4. The problem statement is thorough, and the challenges faced are comprehensively presented.

**Weaknesses:**

See questions.

**Questions:**

1. In Table 1, the row referring to SDformer V2 states that the work uses the SNN-unfriendly SEW-Shortcut operation. However, SDformer V2 actually adopts the MS-Shortcut connection method.

2. For E-Spikeformer, as the authors mentioned, it is indeed difficult to achieve step-by-step computation. However, the paper discusses possible asynchronous implementation schemes. In this case, is it still necessary to incur high costs to obtain an SNN through hybrid training (Conversion + Learning), especially when its performance is inferior to directly trained SNNs? Admittedly, such asynchronous implementations rely on specific neuromorphic chips, but I believe that for SNNs, leveraging neuromorphic hardware is inherently essential to fully realize their advantages.

3. The hybrid learning framework used by the authors relies on A2S, which means it first requires selecting an ANN architecture. What kind of ANN architecture did the authors use in this case?

---

> ### Author Response · Authors · 2025-11-29
> **Response to Reviewer XRYA (1/3)**
>
> **[Question1]**  In Table 1, the row referring to SDformer V2 states that the work uses the SNN-unfriendly SEW-Shortcut operation. However, SDformer V2 actually adopts the MS-Shortcut connection method.
>
> **[Answer for Question1:]**
>
> We apologize for the typo in Table 1. SDFormer V2 actually adopts the MS-Shortcut connection method, so it still satisfies the addition-only property in Table 2. We will correct the typo in Table 1 in the revised manuscript.

---

> > ### Author Response · Authors · 2025-11-29
> > **Continue: Response to Reviewer XRYA (2/3)**
> >
> > **[Question2]** For E-Spikeformer, as the authors mentioned, it is indeed difficult to achieve step-by-step computation. However, the paper discusses possible asynchronous implementation schemes. In this case, is it still necessary to incur high costs to obtain an SNN through hybrid training (Conversion + Learning), especially when its performance is inferior to directly trained SNNs? Admittedly, such asynchronous implementations rely on specific neuromorphic chips, but I believe that for SNNs, leveraging neuromorphic hardware is inherently essential to fully realize their advantages.
> >
> > **[Answer for Question2:]**
> >
> > We agree that on specific neuromorphic chips, E-Spikeformer[1] can leverage asynchronous computation to improve performance and energy efficiency. However, the computation pattern and the data-dependence in SNN itself are `also vital for real-world deployment and latency reduction`. For example, some real-world tasks requires step-by-step pattern, as say in SpikingJelly[2]:
> > > In real-world tasks,..., T can be infinity in some online tasks which requires the network to output Y[t] as soon as possible when it receives X[t],..., In these cases, only the step-by-step pattern can be used.
> >
> > Step-by-step pattern also `enables early exit at intermediate time-steps, which has been demonstrated by previous works [3,4,5], to provide around 33% latency savings even on general-purpose GPU hardware.` Therefore, for scenarios that require a step-by-step pattern, a hybrid training framework with SNN-friendly operators is essential to improve SNN accuracy and efficiency.

---

> > > ### Author Response · Authors · 2025-11-29
> > > **Continue: Response to Reviewer XRYA (3/3)**
> > >
> > > **[Question3]** The hybrid learning framework used by the authors relies on A2S, which means it first requires selecting an ANN architecture. What kind of ANN architecture did the authors use in this case?
> > >
> > > **[Answer for Question3:]**
> > >
> > > We select an ANN architecture similar to those used in traditional A2S methods (e.g., MST [6], SpikeZIP-TF [7]). Specifically, similar to MST, we use Swin Transformer [8] as the ANN backbone for all the datasets. The detailed network architecture is shown in Table A5 in the appendix.
> > >
> > > To replace SNN-unfriendly operators (softmax, LayerNorm, GELU) without harming ANN performance, we introduce a novel activation function, UnifiedClip, whose quantized form is equivalent to an ST-BIF neuron. By substituting these operators with UnifiedClip, we build UniFormer, a unified transformer architecture suitable for both ANN and SNN training.
> > >
> > > **[Reference]**
> > >
> > > [1] Scaling Spike-driven Transformer with Efficient Spike Firing Approximation Training
> > >
> > > [2] SpikingJelly: An open-source machine learning infrastructure platform for spike-based intelligence
> > >
> > > [3] Unleashing the Potential of Spiking Neural Networks with Dynamic Confidence
> > >
> > > [4] Knowing when to stop: Delay-adaptive spiking neural network classifiers with reliability guarantees
> > >
> > > [5] SEENN: Towards Temporal Spiking Early-Exit Neural Networks
> > >
> > > [6] Masked Spiking Transformer
> > >
> > > [7] SpikeZIP-TF: Conversion is All You Need for Transformer-based SNN
> > >
> > > [8] Swin Transformer: Hierarchical Vision Transformer using Shifted Windows

---

### Official Review · Reviewer_h7XN · 2025-10-29

**Soundness:** 2
**Presentation:** 2
**Contribution:** 2
**Rating:** 2
**Confidence:** 5

**Summary:**

This work proposes UniSpike, a hybrid training framework that combines the high-accuracy feature of ANN-to-SNN conversion algorithms and the ultra-low time-step inference feature of direct training algorithms.

**Strengths:**

This paper is well-presented, and I am glad to see that it highlights the difference between layer-by-layer and step-by-step inference, which helps audiences outside the field understand the core distinction between ANNs and SNNs.

**Weaknesses:**

1. Regarding the benefit of "hybrid training". The benefit of ANN-to-SNN conversion is the saved training time, while we can enjoy the high performance brought by the pretrained model in ANN. However, this work requires first finetuning the ANN into a quantized ANN and then conducting another round of finetuning for the converted SNN. According to the Appendix, I found the UniFormer actually requires a total of **620** training epochs, far more than the directly trained models/previous converted SNN benchmarks. Then, what's the actual benefit of this method in terms of training cost?

2. Regarding the effectiveness. Hybrid Training also originates from a pretrained ANN. Performance evaluation should focus more on the performance gap between ANN and SNN, rather than just the performance of the converted SNN. On ImageNet, UniFormer still suffers a -2.47% performance drop, while the ANN benchmarks on the other datasets are not reported. The performance gap is significant with the advancements in the conversion method. For instance, ECMT only suffers from ~-1% performance drop at T=4 in their ICML version; their DCGS achieved up to 1% performance loss in a training-free manner. Actually, UniFormer cannot compete with SOTA directly trained SNN Transformers (QKFormer was actually presented in Neurips 2024, nearly 1 year ago).

3. Regarding energy consumption comparisons of QKFormer, equating possible ternary spike operations with "MAC" operations is inappropriate. In fact, the "SEW shortcut" is currently the only shortcut mechanism supported by existing neuromorphic hardware; on asynchronous hardware, even if the number of input spikes is greater than one, it is still processed as a synaptic operation. Therefore, energy consumption estimates should be based on actual synaptic operation count. In fact, QKFormer has already taken into account the case where the input spike is greater than 1 in their energy consumption estimates. Thus, the authors should instead follow the energy consumption results originally reported in QKFormer and modify the data presented in Table 2 and Table 4 to ensure a fair comparison and avoid potential misleading information.

4. In Table 1, QCFS and MST used BatchNorm instead of LayerNorm. The authors should correct this mistake.

**Questions:**

Please address the points listed in Weakness.

Furthermore, although the authors claimed they proposed the "Unified Clip" operator that can replace softmax, layer normalization, and GeLU operation, I found that the core idea behind "unifying" these operators is still to avoid using them (the original softmax, LN, GeLU operation in the pretrained model must be eliminated before conversion). This is well-adopted in prior works; it's just that the previous work didn't state it in this way. The direct result is a -1.67% performance drop in QANN before the conversion. I think this contribution is somewhat exaggerated, and I would like to ask the authors to clarify this point.

---

> ### Author Response · Authors · 2025-11-29
> **Response to Reviewer h7XN (1/5)**
>
> **[Question1]** Regarding the benefit of "hybrid training". The benefit of ANN-to-SNN conversion is the saved training time, while we can enjoy the high performance brought by the pretrained model in ANN. However, this work requires first finetuning the ANN into a quantized ANN and then conducting another round of finetuning for the converted SNN. According to the Appendix, I found the UniFormer actually requires a total of 620 training epochs, far more than the directly trained models/previous converted SNN benchmarks. Then, what's the actual benefit of this method in terms of training cost?
>
> **[Answer for Question1:]**
>
> We thank the reviewer for raising this insightful question regarding the training cost.
> `The training cost advantage of UniSpike is its much faster convergence.` As shown in Table R2.1, UniSpike reaches 77.00% accuracy within only 10 hours, already outperforming SDFormerV2 (68.06%) and QKFormer (66.14%). This faster convergence stems from UniSpike’s training strategy: it first pre-trains an ANN model, which is substantially faster and more stable to optimize than directly training an SNN.
>
> > Table R2.1: Accuracy of different methods at various training hours. **For UniSpike, 0h-38.5h is ANN training, 38.5h-51.3h is quantization-aware training and 51.3h-65h is SNN fintuning.**
>
> | Accuracy   | 10h   | 20h   | 30h   | 40h   | 50h   | 60h   | 70h   | 124h  |
> | ---------- | ----- | ----- | ----- | ----- | ----- | ----- | ----- | ----- |
> | SDFormerV2[1] | 68.06 | 73.83 | 77.56 | 78.95 | 79.73 | 79.80 | 79.80 | 79.80 |
> | QKFormer[2]   | 66.14 | 71.05 | 72.96 | 74.38 | 76.69 | 77.53 | 78.93 | 84.26 |
> | UniSpike   | 77.00 | 79.63 | 82.15 | 80.95 | 81.78 | 80.16 | 80.83 | 80.83 |
>
> `Another advantage of UniSpike is that it removes the softmax, LayerNorm, and GELU operators with a simple element-wise function: UnifiedClip, reducing the overall training time.` As shown in Table R2.2, compared to the UniFormer with LayerNorm and softmax, UnifiedClip reduces the average training time by 7.6\% on 8$\times$RTX4090, from 70.36 hours to 65.00 hours. The reason is that softmax and LayerNorm need to calculate the mean or variance for each token, which is computationally expensive. In contrast, UnifiedClip is an element-wise operator, simplifying the calculation significantly.
>
> > Table R2.2: Training hours of UniFormer with different operators.
>
> | 8$\times$RTX4090 | Layernom+Softmax | Layernorm | UnifiedClip |
> | ---------------- | ---------------- | --------- | ----- |
> | ANN(300 epoch)   | 41.99h            | 41.23h     | 38.54h |
> | QANN(40 epoch)   | 13.57h            | 13.49h     | 12.80h |
> | SNN(10 epoch)    | 14.80h            | 14.46h     | 13.66h |
> | Total(350 epoch) | 70.36h            | 69.18h     | 65.00h |

---

> ### Author Response · Authors · 2025-11-29
> **Continue: Response to Reviewer h7XN (2/5)**
>
> **[Question2]** Regarding the effectiveness. Hybrid Training also originates from a pretrained ANN. Performance evaluation should focus more on the performance gap between ANN and SNN, rather than just the performance of the converted SNN. On ImageNet, UniFormer still suffers a -2.47% performance drop, while the ANN benchmarks on the other datasets are not reported. The performance gap is significant with the advancements in the conversion method. For instance, ECMT only suffers from ~-1% performance drop at T=4 in their ICML version; their DCGS achieved up to 1% performance loss in a training-free manner. Actually, UniFormer cannot compete with SOTA directly trained SNN Transformers (QKFormer was actually presented in Neurips 2024, nearly 1 year ago).
>
> **[Answer for Question2:]**
>
> Regarding effectiveness, UniSpike addresses two key problems in SNN training:
>
> **Problem 1 & Solution:** For the conversion methods, they still suffer from the significant accuracy loss at extremely low time-steps (e.g., 4 time-steps). As shown in Table R2.3, with similar model parameters, the state-of-the-art conversion method ECMT suffers from a 10.79% accuracy loss, `demonstrating that improving accuracy at ultra-low time-steps in converted SNNs is still challenging.` In contrast, with SNN finetuning, UniSpike further reduces the accuracy loss, from 21.08% to 2.47%, achieving the SOTA performance at 4 time-steps.
>
> > Table R2.3: Comparison of accuracy drops with advanced conversion methods. * are reproduced by us. Data are taken from the original papers. F.T. denotes SNN fintuning.
>
> |               | SZIP-TF[4]*     | MBE[5] | ECMT[6] | UniSpike(w/o F.T.) | UniSpike  |
> | ------------- | --------------- | ------ | ------- |-------------------  | --------- |
> | Architecture  | VIT-B           | VIT-B  | VIT-B   |   UniFormer         | UniFormer |
> | Parameter(M)  | 86.6            | 86.6   | 86.6    |   66.5              | 66.5      |
> | Time-step     | 4               | 8      | 4       |   4                 | 4         |
> | ANN Accuracy(%)  | 83.75           | 83.44  | 80.77   |   83.30             | 83.30     |
> | SNN Accuracy(%)  | 0.10             | 0.12   | 69.98   |   62.22             | 80.83     |
> | Accuracy Loss(%) | 83.65           | 83.32  | 10.79   |   21.08             | **2.47**  |
>
> **Problem 2 & Solution:** For the directly trained SNN transformers, spike-driven methods (*e.g.,* Spike-driven transformer V2) with spike self-attention (SSA) and MS shortcut (also called "pre-activation residual shortcut" in QKFormer) cannot achieve high accuracy comparable to ANN. Other learning-enhancing operators, such as the SEW-Shortcut in QKFormer[2] and MSVIT[3], significantly increase the number of spikes, which hinders the efficiency advantages of SNNs. `Maintaining the high energy-efficiency feature while achieving high accuracy is a challenge in directly trained SNN transformers.` UniSpike proposes a spike-driven spiking transformer by removing the SNN-unfriendly operators and improving its accuracy witha  hybrid training strategy. As shown in Table R2.4, we compare UniSpike with SOTA directly trained SNNs. Thanks to the highly energy-efficient architecture (*i.e,* UniFormer), UniSpike achieves the highest accuracy (80.83%) with similar energy consumption (15mJ ~ 18mJ).
>
> > Table R2.4: Energy-Accuracy comparison with SOTA direct learning methods. Data from the original paper.
>
> |             | SDformer V2[1]  | QKFormer[2] | MSVIT[3] | UniSpike    |
> | ----------- | --------------  | ----------- | -------- | ----------- |
> | Energy(mJ)  | 16.70           | 15.13       | 16.65    | 18.0        |
> | Accuracy(%) | 74.10           | 78.80       | 80.09    | 80.83       |

---

> ### Author Response · Authors · 2025-11-29
> **Continue: Response to Reviewer h7XN (3/5)**
>
> **[Question3]**  Regarding energy consumption comparisons of QKFormer, equating possible ternary spike operations with "MAC" operations is inappropriate. In fact, the "SEW shortcut" is currently the only shortcut mechanism supported by existing neuromorphic hardware; on asynchronous hardware, even if the number of input spikes is greater than one, it is still processed as a synaptic operation. Therefore, energy consumption estimates should be based on actual synaptic operation count. In fact, QKFormer has already taken into account the case where the input spike is greater than 1 in their energy consumption estimates. Thus, the authors should instead follow the energy consumption results originally reported in QKFormer and modify the data presented in Table 2 and Table 4 to ensure a fair comparison and avoid potential misleading information.
>
> **[Answer for Question3:]**
>
> We sincerely thank the reviewer for the valuable comment regarding energy consumption, and we would like to clarify the misunderstanding related to the SEW-Residual Addition. As shown in the QKFormer block pseudocode below and in Table R2.5, the inputs to mlp1/attn2/…/mlpL `are not strictly binary or ternary activations; instead, they take on multiple discrete values.` This is because SEW-Residual Addition performs an identity bypass without a neuron layer, and the spikes are accumulated at each block, causing a non-binary or non-ternary feature map in deep blocks. Therefore, we consider the operators after SEW-ResNet as "MAC" operations in Table 4.
> ```python
> # The Pseudocode in QKFormer Blocks
> x = x + self.attn1(x) # BLOCK 1
> x = x + self.mlp1(x) # BLOCK 1
>
> x = x + self.attn2(x) # BLOCK 2
> x = x + self.mlp2(x) # BLOCK 2
> ...
> x = x + self.attnL(x) # BLOCK L
> x = x + self.mlpL(x) # BLOCK L
> ```
> > Table R2.5: The value set of spike activations in SNNs equipped with SEW-Residual Addition
>
> |       | Input           | Residual Input | Output        |
> | ----- | --------------- | -------------- | ------------- |
> | attn1 | {0,1}           | -              | {0,1}         |
> | Add   | {0,1}           | {0,1}          | {0,1,2}       |
> | mlp1  | {0,1,2}         | -              | {0,1}         |
> | Add   | {0,1}           | {0,1,2}        | {0,1,2,3}     |
> | attn2 | {0,1,2,3}       | -              | {0,1}         |
> | Add   | {0,1}           | {0,1,2,3}      | {0,1,2,3,4}   |
> | mlp2  | {0,1,2,3,4}     | -              | {0,1}         |
> | Add   | {0,1}           | {0,1,2,3,4}    | {0,1,2,3,4,5} |
> | ....  | ....            | ....           | ....          |
> | mlpL  | {0,1,2,..., 2L} | -              | {0,1}         |
>
> We follow the reviewer’s suggestion and present the corresponding #SOP breakdown in Table R2.6. As shown in Table R2.6, we compare the #SOP breakdown of QKFormer and UniFormer. Since SEW Residual increases the number of spikes significantly, the #SOP of Query, Key, Value, and FC1 are on average 1.83$\times$ larger than UniFormer, demonstrating UniFormer is an energy-efficient SNN architecture.
>
> > Table R2.6: The #SOP breakdown of QKFormer and UniFormer. * reproduced by ourselves.
>
> |                 | QKFormer-10-768(65.0M)* | UniFormer(66.5M) |
> | --------------- | ----------------------- | ---------------- |
> | Query           | 3.03  GACs               | 1.89 GACs        |
> | Key             | 3.69  GACs               | 1.89 GACs        |
> | Value           | 2.52 GACs               | 1.89 GACs        |
> | Attn            | 0.24 GACs               | 1.59 GACs        |
> | Y               | 0.88 GACs               | 0.74 GACs        |
> | FC1             | 13.87 GACs               | 6.95 GACs        |
> | FC2             | 1.25 GACs               | 0.49 GACs        |
> | Patch Embedding | 5.84 GACs               | 1.72 GACs        |
> | Patch Merging   | 12.87 GACs               | 3.30 GACs        |
> | Total           | 44.17 GACs              | 19.97 GACs       |

---

> ### Author Response · Authors · 2025-11-29
> **Continue: Response to Reviewer h7XN (4/5)**
>
> **[Question4]** Furthermore, although the authors claimed they proposed the "Unified Clip" operator that can replace softmax, layer normalization, and GeLU operation, I found that the core idea behind "unifying" these operators is still to avoid using them (the original softmax, LN, GeLU operation in the pretrained model must be eliminated before conversion). This is well-adopted in prior works; it's just that the previous work didn't state it in this way. The direct result is a -1.67% performance drop in QANN before the conversion. I think this contribution is somewhat exaggerated, and I would like to ask the authors to clarify this point.
>
> **[Answer for Question4:]**
>
> We sincerely thank the reviewer for providing us with the opportunity to further clarify the motivation and contributions of our work. UniSpike targets an SNN with **high accuracy and high energy efficiency**, which has not been well achieved in previous SNN works.
>
> In previous conversion works [4,5,6], the softmax, LayerNorm and GELU are inevitable components in SNN, since the ANN and SNN structure must be the same. However, these operators required dense MAC operations[1,7] and are incompatible with neuromorphic hardware [8,9], making SNNs inefficient. Therefore, UniSpike proposes UniFormer, an ANN architecture tailored for SNNs that `attains high accuracy without relying on these inefficient operators, instead of passively removing them.`
>
> In directly trained SNN transformers, the softmax, LayerNorm, and GELU are indeed eliminated as the reviewer mentioned. However, as displayed in Table R2.7, directly training an SNN without SEW-Residual and these unfriendly operators suffers from low accuracy[1,7]. To further improve accuracy, UniSpike `uses a hybrid training method to provide a good weight initialization for SNN finetuning rather than training from scratch`. Although UniSpike shows a performance drop compared with the original ANN, as shown in Table R2.7, it still surpasses SOTA SNN (e.g., SDformer V2 [1]) that do not rely on SEW-Residual or other unfriendly operators. This demonstrates the contribution of UniSpike in achieving SNN with high accuracy while maintaining high energy efficiency.
>
> > Table R2.7: Comparison with Spiking Transformers without SEW-ResNet and softmax, LayerNorm and GELU operators.
>
> | Methods                     | Parameter(M) | Energy(mJ) | Time-Step | Accuracy(%) |
> | --------------------------- | ------------ | ---------- | --------- | ----------- |
> | SpikingFormer               | 66.34        | 13.68      | 4         | 75.85       |
> | Spike-Driven Transformer    | 66.34        | 6.09       | 4         | 77.07       |
> | Spike-Driven Transformer V2 | 55.4         | 52.4       | 4         | 80.00       |
> | UniSpike                    | 66.5         | 18.0       | 4         | 80.83       |

---

> > ### Author Response · Authors · 2025-11-29
> > **Continue: Response to Reviewer h7XN (5/5)**
> >
> > **[Question5]** In Table 1, QCFS and MST used BatchNorm instead of LayerNorm. The authors should correct this mistake.
> >
> > **[Answer for Question5:]**
> >
> > Thanks for your careful reading. We will correct this in the revised manuscript.
> >
> > **[Reference]**
> >
> > [1] Spike-driven Transformer V2: Meta Spiking Neural Network Architecture Inspiring the Design of Next-generation Neuromorphic Chips
> >
> > [2] QKFormer: Hierarchical Spiking Transformer using Q-K Attention
> >
> > [3] MSVIT: Improving Spiking Vision Transformer Using Multi-scale Attention Fusion
> >
> > [4] SpikeZIP-TF: Conversion is All You Need for Transformer-based SNN
> >
> > [5] Training-Free ANN-to-SNN Conversion for High-Performance Spiking Transformer
> >
> > [6] Towards High-performance Spiking Transformers from ANN to SNN Conversion
> >
> > [7] Spike-driven transformer
> >
> > [8] Sorbet: A Neuromorphic Hardware-Compatible Transformer-Based Spiking Language Model
> >
> > [9] TrueNorth: Design and Tool Flow of a 65 mW 1 Million Neuron Programmable Neurosynaptic Chip

---

### Official Review · Reviewer_f9dh · 2025-10-31

**Soundness:** 3
**Presentation:** 3
**Contribution:** 3
**Rating:** 6
**Confidence:** 4

**Summary:**

This paper proposes UNISPIKE, a hybrid training framework for Spiking Neural Networks (SNNs) that combines the high accuracy of ANN-to-SNN conversion (A2S) and the ultra-low time-step inference of direct training (DT). It introduces UnifiedClip to replace SNN-unfriendly operators (e.g., softmax, layernorm, GeLU) and builds UniFormer, an addition-only, step-by-step inference transformer. Experiments on ImageNet, CIFAR datasets, and CIFAR10-DVS show UNISPIKE outperforms SOTA SNN algorithms in accuracy, energy efficiency, and latency.

**Strengths:**

1.UNISPIKE effectively merges A2S and DT advantages, addressing the trade-off between SNN accuracy and inference time-steps, a key challenge in SNN research.
2.UnifiedClip solves SNN-unfriendly operator issues without degrading ANN accuracy, enabling the novel UniFormer with energy-saving addition-only computation.
3.Comprehensive experiments on multiple datasets (static and neuromorphic) demonstrate UNISPIKE’s superiority over SOTA methods in accuracy, energy use, and latency.

**Weaknesses:**

1. I am confused that since the paper proposes UnifiedClip and it is equivalent to ST-BIF neurons, what are the differences between UnifiedClip and ST-BIF neurons, or what advantages does UnifiedClip have over ST-BIF neurons?It seems that UnifiedClip (UClip) is a operator of Artificial Neural Networks (ANNs), and due to its equivalence to ST-BIF neurons, the accuracy loss during the ANN-to-SNN (A2S) conversion is relatively small?

2.In Table 3, UniSpike with UniFormer-T1 uses more time-steps than QKFormer (6 > 4) and achieves higher accuracy than QKFormer on CIFAR10 and CIFAR100. This part of the comparison seems to fail to fully reflect its advantages. How about the accuracy of UniSpike with timestep=4?

**Questions:**

See Weaknesses.

---

> ### Author Response · Authors · 2025-11-29
> **Response to Reviewer f9dh (1/2)**
>
> **[Question1]** I am confused that since the paper proposes UnifiedClip and it is equivalent to ST-BIF neurons, what are the differences between UnifiedClip and ST-BIF neurons, or what advantages does UnifiedClip have over ST-BIF neurons?It seems that UnifiedClip (UClip) is a operator of Artificial Neural Networks (ANNs), and due to its equivalence to ST-BIF neurons, the accuracy loss during the ANN-to-SNN (A2S) conversion is relatively small?
>
> **[Answer for Question1:]**
>
> One of the contributions of UnifiedClip over ST-BIF neurons is that it remains equivalent to ST-BIF neurons while providing an accurate gradient when training. This `enables high-accuracy ANN training and offers good weight initialization for SNN finetuning` in UniSpike. As shown in Table R1.1, directly training algorithms [1] suffers from slow convergence and lower accuracy compared to the ANN (8.94% lower accuracy with 10 hours of training). The reason is that a spiking neuron is non-differentiable and the surrogate gradient is inaccurate.
>
> > Table R1.1: Test accuracy comparison between SOTA Spike-Driven SNN and ANN with UnifiedClip during training from scrach on ImageNet. GPU: 8$\times$RTX4090. **For UniSpike, 0h-38.5h is ANN training, 38.5h-51.3h is quantization-aware training and 51.3h-65h is SNN fintuning.**
>
> | Test Accuracy   | Param.(M)| 10h   | 20h   | 30h   | 40h   | 50h   | 60h   |
> | ---------- | -----| ----- | ----- | ----- | ----- | ----- | ----- |
> | SNN (SDFormerV2[1]) | 55.4 | 68.06 | 73.83 | 77.56 | 78.95 | 79.73 | 79.80 |
> | ANN (UnifiedCLIP)  | 66.5 | 77.00 | 79.63 | 82.15 | 83.30 | 83.30 | 83.30 |
>
> To solve the problem, we propose UnifiedClip, `which enables a hybrid training method to improve SNN accuracy`. We first train a high-accuracy ANN with UnifiedClip, then convert the ANN to an SNN with a good weight initialization. As shown in Table R1.2, after conversion, UniFormer achieves 62.22% accuracy at 4 time-steps, demonstrating the effectiveness of weight initialization. Finally, we finetune the SNN to obtain an SNN with higher accuracy(80.83%) than training SNN from scratch (79.8%), as shown in Table R1.1.
>
> > Table R1.2: Accuracy before and after SNN finetuning.
>
> |               | UniSpike(w/o F.T.) | UniSpike  |
> | ------------- | ------------------- | --------- |
> | Architecture  |UniFormer          | UniFormer |
> | Parameter(M)  | 66.5               | 66.5      |
> | Time-step     |   4                | 4         |
> | SNN Accuracy(%) | 62.22             | 80.83     |

---

> > ### Author Response · Authors · 2025-11-29
> > **Continue: Response to Reviewer f9dh (2/2)**
> >
> > **[Question2]** In Table 3, UniSpike with UniFormer-T1 uses more time-steps than QKFormer (6 > 4) and achieves higher accuracy than QKFormer on CIFAR10 and CIFAR100. This part of the comparison seems to fail to fully reflect its advantages. How about the accuracy of UniSpike with timestep=4?
> >
> > **[Answer for Question2:]**
> >
> > We provide the accuracy of UniFormer-T1 on CIFAR-10 and CIFAR-100 at 4 time-steps as requested by the reviewer in Table R1.3. At the same time-step, UniSpike still achieves the SOTA performance, outperforming QKFormer by 0.26% on CIFAR-10 and 0.68% on CIFAR-100.
> >
> > > Table R1.3: Experimental results on CIFAR-10 and CIFAR-100.
> >
> > |   Methods         | Category |     Arch.    | Param. | T |  CIFAR-10 | CIFAR-100 |
> > |:-----------------:|:--------:|:------------:|:------:|:-:|:-----:|:-----:|
> > | Spikingformer[2]     |    SNN   |     4-384    |  9.32  | 4 | 95.81 | 78.21 |
> > |   S-Transformer[3]   |    SNN   |     4-384    | 10.28  | 4 | 95.60 | 78.40 |
> > |     QKFormer[4]      |    SNN   |     4-384    |  6.74  | 4 | 96.18 | 81.15 |
> > | UniSpike (ours)   |    SNN   | UniFormer-T1 |  4.50  | 4 |  **96.44**  |**81.83**|
> >
> > **[Reference]**
> >
> > [1] Spike-driven Transformer V2: Meta Spiking Neural Network Architecture Inspiring the Design of Next-generation Neuromorphic Chips
> >
> > [2] Spikingformer: A Key Foundation Model for Spiking Neural Networks
> >
> > [3] Spike-driven transformer
> >
> > [4] QKFormer: Hierarchical Spiking Transformer using Q-K Attention

---

### Author Response · Authors · 2025-11-29
**Response to Area Chair and All Reviewers**

Dear Area Chair and All Reviewers:

We fully understand the additional workload caused by the recent OpenReview bug and sincerely appreciate the program committee's continued efforts to maintaining the integrity of the ICLR review process.

We thank all reviewers for their constructive and insightful feedback. In the revised manuscript, we have made the following key improvements and clarification:

1. Contribution and Component Clarification
- Clarify the advantages of UnifiedClip over ST-BIF neuron.
- Further exhibit the UniSpike benefits regarding to training cost, task accuracy and energy efficiency.
- Clarify the motivation of UnifiedClip that replaces the SNN-unfriendly operators including Softmax, LayerNorm and GeLU.

2. Extended Evaluation and Analysis
- Add a comparison of accuracy over training time between UniSpike and other methods.
- Add the CIFAR-10/CIFAR-100 accuracy of UniSpike at 4 time-steps
- Provide a comparison between UniSpike and the latest conversion-based SNNs, as well as SOTA directly trained SNNs.
- Analyze the training cost of UnifiedClip compared to LayerNorm and Softmax.

3. Neuromorphic Hardware and Real-world Consideration
- UniSpike is still essential for step-by-step SNNs at neuromorphic hardware and real-world tasks.

Please see our detailed responses below each review. If you have any questions or would like additional information, we are happy to provide further details. Thank you again for your time and service to the community.

Best regards,

The Authors

---

### Meta-Review · Area_Chair_LhRL · 2025-12-26

**Summary:**

This paper introduces UniSpike, a new technique to train spiking neural networks (SNNs). SNNs mimic how real brain neurons fire in short bursts (spikes) instead of using MAC operations in regular AI models. The goal is to make SNNs work better and faster while consuming less power, benefiting tiny devices or energy-saving AI. UniSpike mixes two training approaches: first converting a regular AI model (ANN) into an SNN, then fine-tuning the SNN.

UniSpike has two main components:
1. UnifiedClip: A way to replace or remove certain math operations (like Softmax, LayerNorm, and GeLU).
2. UniFormer: A simplified transformer design that only uses addition.

The authors claimed UniSpike achieves better accuracy than other SNNs when trained with very few epochs. It used less energy. They tested it on image datasets like ImageNet and CIFAR. Most reviewers thought the paper has a lot of problems. Many reviewers found it confusing, overstated, and not convincingly better than existing methods.

**Reviewer Concerns:**

Main Concerns of Most Reviewers

1. The Training Is Slow. The selling point of this paper is that UniSpike combines the best of two training styles for faster and better results. But reviewers said: (1) It takes hundreds of training epochs — sometimes over 600; (2) It needs more than 60 GPU hours; and thus (3) This is actually more expensive than the older methods. So the claim that it saves time or effort is not valid.

2. UnifiedClip Is NOT New. UnifiedClip is one of the contributions of this paper. But reviewers said it was just removing or avoiding those operations — something many earlier SNN papers already do. The regular AI models have lost some accuracy before conversion.

3. Comparisons are NOT Fair. Reviewers found problems: The authors sometimes compare accuracy at different training epochs. Energy savings are based on assumptions that depend on specific hardware and might not be realistic. Some prior methods they compare against were described incorrectly at first (e.g., wrong types of shortcuts or normalization), and fixes were done after reviewers complained.

4. NOT Clearly Better than Existing Methods. UniSpike does NOT always compare the latest or strongest ones. When all conditions are compared fairly (same speed, same setup), UniSpike does NOT always win.

Reviewer-Specific Questions:

1. One reviewer (h7XN) was very negative, pointing out high training costs, misleading energy claims, overhyped UnifiedClip, and wrong baseline details. These issues were NOT fixed even after the authors responded.

2. Another (ZHzt) worried about a possible rule-breaking formatting issue that could affect fairness.

3. Others (f9dh, XRYA) asked for basic clarifications because the paper was hard to understand at first, especially on fair comparisons.


**I do NOT think the authors answered the questions from reviewer h7XN well.** The tables in the authors' rebuttal are problematic. Table R2.1 shows that UniSpike does not need more than 30h training. Table 2.4 shows UniSpike has higher accuracy yet larger energy consumption than QKFormer. Table 2.6 shows UniSpike requires less operations than QKFormer. For me, these three tables have conflicts. I personally do not think the training energy or cost matters too much. More attentions should be paid on the inference parts, which will be invoked for billions of times. Because the higher accuracy of UniSpike comes with the cost of higher energy. This is not a clear win.

**Reviewer Scores:**

I think the scores of reviewers are reasonable.

---

### Decision · Program_Chairs · 2026-01-26

Reject